# Changes in Use of Blood Cultures in a COVID-19-Dedicated Tertiary Hospital

**DOI:** 10.3390/antibiotics11121694

**Published:** 2022-11-24

**Authors:** Alina-Ioana Andrei, Gabriel-Adrian Popescu, Mona Argentina Popoiu, Alexandru Mihai, Daniela Tălăpan

**Affiliations:** 1Faculty of Medicine, “Carol Davila” University of Medicine and Pharmacy, 050474 Bucharest, Romania; 2“Prof. Dr. Matei Balș” National Institute of Infectious Diseases, 021105 Bucharest, Romania

**Keywords:** blood culture, contamination rate, COVID-19

## Abstract

Blood cultures should be collected within an hour in the setting of sepsis/septic shock. The contamination rate should be below 3%. Worldwide reports have described an increase in blood contamination rates during the COVID-19 pandemic. We performed a retrospective analysis of the blood cultures collected during a 10-month period (March–December 2020) at NIID “Prof. Dr. Matei Balș”. The results were compared with data from the pre-pandemic period (March–December 2016) and with the existing data in the literature. During the pandemic, there was a significant decrease in the number of blood cultures collected (1274 blood cultures in 2020 vs. 5399 in 2016). The contamination rate was higher in 2020 (11.7%) compared to 2016 (8.2%), *p* < 0.001. The rate of infectious episodes in which the etiological agent was identified was constant: 11% in 2020 versus 11.9% in 2016, *p* = 0.479, but there were fewer invasive bacterial/fungal infections: 0.95/1000 patient days in 2020 vs. 2.39/1000 patient days in 2016, *p* < 0.001. We observed a change in the species distribution. The Gram-negative isolate’s proportion increased from 50.6% to 63.1% and the gram-positive isolate’s proportion decreased from 31.8% to 19%. Collection of a low number of blood cultures and a high contamination rate was identified in our clinic. In order to improve the usefulness of blood cultures as a diagnostic method, at least two sets should be collected in aseptic conditions.

## 1. Introduction

Invasive infections are life-threatening conditions and a diagnosis should be established as soon as possible, and the appropriate antimicrobial treatment should be started immediately.

Blood cultures are an important tool in the diagnosis of invasive infections. According to the latest surviving sepsis campaign, recommendations and blood cultures should be collected within the first hour when sepsis/septic shock is suspected [1]. In the latest definition, sepsis is defined as a life-threatening organ dysfunction caused by a dysregulated host response to infection [2].

In addition to establishing the etiological diagnosis, blood cultures can also help clinicians decide the most appropriate empiric antimicrobial therapy via the Gram-stained smear (the first examination performed once blood cultures are positive) and later the specific one, after antimicrobial susceptibility testing is performed.

They are also useful tools for determining when to stop antimicrobial therapy [3].

Currently, there is no consensus regarding blood culture collection indications. Shapiro et al. [4] postulated that blood cultures should be collected in the presence of more than one major criterion or at least two minor criteria. The major criteria proposed were suspected endocarditis, a temperature ≥ 39.4 °C, and the presence of indwelling vascular advice. The minor criteria consisted of an age > 65, a temperature of 38.3–39 °C, chills, vomiting, systolic blood pressure ≤ 90 mmHg, white cell count > 18, bands > 5% platelets < 150, and creatinine > 200 mg/dL.

Recently, Otani et al. [5] published a study that aimed to determine which criteria should be better used to detect patients who need blood cultures: SIRS, qSOFA, or Shapiro’s criteria. The retrospective analysis included 986 patients and concluded that Shapiro’s clinical prediction rule had the best sensitivity, whereas patients with sepsis and positive qSOFA score had better specificity.

In order to perform an optimal diagnosis, Baron et al. [6] recommended that the number of blood cultures collected in a hospital designated for the treatment of acute infections should range between 103–188/1000 patient days.

Regardless of the collection indications, in order to have a higher chance to make a correct diagnosis, cultures should be obtained prior to the initiation of any antimicrobial therapy [7], in a proper quantity (40–60 mL of blood/blood culture set) and a number of sets (2–3 sets of blood cultures, each set of blood cultures consisting of two bottles: one with aerobic medium and the other one with anaerobic medium) [8,9]. Of course, the aseptic conditions should be respected during the collection procedure.

Non-compliance with the collection protocol can cause blood culture contamination. This event involves introducing into the blood culture vial a germ that does not come from the patient’s blood but from the environment, the patient’s skin, or from the staff who collected it. The most frequent contaminant agents recovered are the coagulase-negative staphylococci, *Corynebacterium* spp., *Bacillus* spp. other than anthracis, and *Propionibacterium acnes* [10,11].

Blood culture contamination can lead to false diagnosis, increased hospitalization length, increase in antimicrobial usage, thus promoting *Clostridioides difficile* infection and excess cost.

These factors promoted a high variability regarding the identification rate in bacteremia that ranged in the published data from 4% to 7% [12,13].

Published data worldwide retrieved a large range of contaminated blood cultures, between 0.6% [10] and over 10% [14]. Even if this is not a desirable event, blood culture contamination is a reality and, according to current guidelines, its rate should be below 3% and ideally less than 1% [15,16].

In order to reduce the blood culture contamination rate, a wide range of strategies were proposed, such as the usage of alcohol-based antiseptics, the delegation of phlebotomy teams for blood culture collection, special blood culture collection kits, diversion of the first portion of blood, and the usage of sterile gloves together with educational programs for the medical personnel regarding blood culture usage and collection [13,17,18,19,20,21].

COVID-19 is a respiratory pathology produced by the SARS-CoV-2 virus that causes fever and respiratory tract symptoms (runny nose, cough, and laryngitis) that can evolve into respiratory failure and multiple organ failure [22,23].

The pathogenesis consists of three stages: viral multiplication, pneumonic, and the immune-inflammatory phase, in which fever and pneumonia can be present. A minority of patients can develop a cytokine storm that can determine a severe form of the disease [24].

In this context, is difficult to differentiate a fever of viral origin from that of bacterial origin based only on clinical and non-specific laboratory work. Patients with severe forms of COVID-19 can meet the sepsis criteria [2].

Although primary bloodstream bacterial/fungal COVID-19 co-infections are very uncommon, secondary bloodstream infections in inpatients can occur, especially in patients admitted to the intensive care unit [25]. This requires increased attention in order to establish the correct etiological diagnosis in the ICU.

Recent worldwide reports have described an increase in blood contamination rates during the COVID-19 pandemic [26,27,28,29].

The aim of our study is to evaluate the changes in the blood cultures usage, contamination rate, frequency, and distribution of the microorganisms isolated from blood cultures collected during the first 10 months of the COVID-19 pandemic in comparison to the pre-pandemic period. The study was focused on the blood collection process, contamination rate, frequency, and distribution of the microorganisms isolated.

## 2. Results

### 2.1. Statistics in the COVID-19 Group

During the study period, 1274 blood culture sets were collected from 782 infectious episodes, with a mean of 1.67 ± 1,16 blood culture sets/infectious episode. The number of blood cultures collected per 1000 patient days was 14.32. From the total collected blood cultures sets, 149 were considered contaminated, which led to a contamination rate of 11.7% (CI95%, 10.1–13.6%), which was significantly higher than the recommended limit of 3%, *p* < 0.0001. The main contaminants isolated were the coagulase-negative staphylococci (91.3%), followed by Propionibacterium acnes (4.7%), *Corynebacterium* spp. (1.34%), and others (2.66%). Blood cultures yielded real pathogens for 84 infectious episodes (11%), mostly from the intensive care unit (53 episodes). The most common pathogens isolated were Acinetobacter baumannii (22 cases, 26.2%), Klebsiella pneumoniae (16 cases, 19%), Staphylococcus aureus (9 cases, 10.7%), and Pseudomonas aeruginosa (6 cases, 7.1%) Table 1.

### 2.2. Statistics in the Pre-COVID-19 Group

Between March and December of 2016, a number of 5399 blood culture sets were collected from patients with 2865 infectious episodes, leading to a mean of 1.91 ± 1.4 blood culture sets/infectious episode. The number of blood cultures collected per 1000 patient days was 38.36, less than the recommended number of collected blood cultures [6]. The contamination rate identified was 8.2% (441 contaminated sets/5399 sets collected), also higher than the recommended limit (*p* = 0.025). Coagulase-negative staphylococci were the most common isolated contaminants (71.7%), followed by *Corynebacterium* spp. (7.3%), *Bacillus* spp. (6.2%), *Micrococcus* spp. (2.9%), Propionibacterium acnes (0.9%), and others (11.1%). Clinically relevant growth was observed in 336 infectious episodes (11.9%), with the main pathogens isolated being Escherichia coli (95 cases, 28.3%), Staphylococcus aureus (71 cases, 21.1%), and Klebsiella pneumoniae (35 cases, 10.4%) Table 1.

The differences between 2020 and 2016 are listed in Table 2 and Table 3.

There was a significant decrease in the number of blood cultures collected during the pandemic (1274 blood cultures in 2020 vs. 5399 in 2016). The contamination rate was significantly higher in 2020 (11.7%) than in 2016 (8.2%).

The rate of infectious episodes in which the etiological agent was identified by blood cultures was relatively constant at 11.9% versus 11%, *p* = 0.479, but there were fewer invasive bacterial/fungal infections: 0.95/1000 patient days vs. 2.39/1000 patient days, and we observed a change in the species distribution: from 31.8% gram-positive isolates to 19%, z score = 2.31, *p* = 0.021.

The number of blood culture sets/infectious episodes decreased, but they were modified in a divergent manner in the ICU (increased) versus the infectious diseases wards (decreased).

## 3. Discussion

During the pandemic period, a decrease in the number of blood culture sets collected was identified in our clinic. The number of blood cultures/1000 patient days was lower even than the range of 103–188/1000 patient days recommended by Baron et al. [6] in the pre-pandemic period. This level is not applicable in the conditions where almost all the patients that were hospitalized in 2020 had COVID-19, a viral infection with a low rate of associated bacterial infections. The level in 2020 is significantly lower compared to 2016, in accordance with this evolution from a predominantly bacterial infectious pathology in the pre-pandemic period to an almost exclusively viral one in 2020.

This finding was similar to data reported in a retrospective study from a tertiary hospital in Spain that detected a 22.7% decrease in the number of blood cultures collected in the pre-covid vs. covid period (6541 cultures collected in 2019 vs. 5313 in 2020) [30], but was in contrast with another study from a multicenter network of New York hospitals that reported an increase in the blood culture collection rate during the pandemic [31]. Possible reasons for the low number of blood cultures collected may be the low number of patients admitted compared to the pre-covid period and the fact that our hospital was dedicated almost exclusively to the management of COVID-19 patients. It was an effect of shifting from the care of patients with infections dominated by a bacterial etiology to patients with SARS-CoV-2 infection, where a minority of bacterial coinfections/superinfections are found. Additionally, the number of collected sets/infectious episodes was low.

The formal indications and practical attitude of the blood culture collection in our hospital were not changed between 2016 and 2020; these facts are supported by the constant true positive rate of blood cultures. If the blood cultures were collected in patients with a high suspicion of bacterial infection, it would have led to an increase in the rate of true positive cultures.

Our study also revealed that when blood cultures were collected, the clinician frequently ordered a single set of blood cultures, maybe two sets, thus decreasing the chances of discovering the etiological agent and increasing the proportion of episodes with an uncertain significance of the isolated microorganism, a real pathogen or a contaminant. In more than half of the infectious episodes, only one set of blood cultures was collected, and the share of these events increased significantly in 2020 compared to 2016. If, for ICU patients, the rate of this situation decreased instead in the infectious disease wards, the collection of a single blood culture set was significantly more common in 2020 than in 2016.

The majority of the collected sets came from the intensive care unit, most likely due to the fact that the rate of bacterial infections is higher, being favored by the severe immunosuppressive state of the patients admitted in this department (produced by both associated diseases and the immunosuppressive therapies administered) and the high rate of bacterial colonization, especially in the setting of orotracheal intubation.

Additionally, blood cultures yielded true pathogens in a smaller number of cases in our hospital. This finding is in accordance with others described in the literature and may be an indication that bloodstream co-infections in COVID-19 are less frequent, but the overall real positive rate of blood cultures was similar in both studied periods. Considering that the majority of the real pathogens were isolated from ICU patients, it can be assumed that COVID-19 bacteremia develops as a secondary event.

At the same time, there was a change in the profile of the microorganisms isolated from blood cultures, the main species detected during COVID-19 being well-known ICU-bacterial species such as *Acinetobacter baumannii*, *Klebsiella pneumoniae* and *Pseudomonas aeruginosa*. In contrast, before the pandemic, blood cultures revealed a diversity of bacteria such as *Escherichia coli*, *Staphylococcus aureus*, *Klebsiella pneumoniae*, *Streptococcus pneumoniae*, *Enterococcus* spp. Likewise, Bayo et al. [30] found a significant increase in hospital-acquired bacteria, from 30.5% during the pre-covid period to 95.5% during the pandemic (*p* < 0.001). The explanation would be that the chance of developing bacteremia in COVID-19 patients is greater as the duration of hospitalization increases, especially in the ICU.

Blood culture contamination continues to be a major problem and can be associated with undesirable events. Similar to data reported in recent studies by Bayo et al. [30] and Ohki [26], the contamination rate in the COVID-19 group was higher than the one before the pandemic (12.3 % vs. 5.7%, *p* < 0.001 and 6.1% vs. 3.7%, *p* = 0.015). As described in other studies [30,31], the coagulase-negative staphylococci were the main contaminants. The reason why the contamination rate is higher in COVID-19 times is not known, but it might be due to the stressful working environment, special working conditions, and the personal protective equipment which can make the collection process more difficult.

In order to decrease the contamination rate, it is necessary to better select the cases in which to collect blood cultures, practicing collection skills in these new working conditions together with improving the knowledge of the medical staff responsible for blood culture collection. At the same time, it is necessary to practice the collection steps, especially in the difficult working conditions encountered during the pandemic.

Although there has been an improvement in the number of blood cultures collected/infectious episodes in the ICU, the percentage of patients with only one set collected remains extremely high. The situation worsened in the infectious diseases wards, with a decrease in the average of blood cultures collected per infectious episode and an increase in the percentage of patients with a single set collected. It is possible that the collection difficulties previously mentioned and the exclusion of an infection associated with COVID-19 based on clinical, biological, and imaging data before the collection of other sets may have contributed to this situation. However, apparently, the principle of the initial collection of two sets of blood cultures has been widely ignored.

### Study Limitations

Our study has limitations. First, we were not able to make a very clear distinction between the real positive bacteria and contaminating germs in all the cases because of the lack of data regarding all the patients studied.

Second, the evaluation of the data recorded in 2021 could highlight improvements regarding this situation, together with the care of a larger number of COVID-19 patients.

Although it would have been useful to analyze the data on changes in antibiotic consumption between the two periods in order to find out if fewer invasive infections resulted in lower antibiotic use and an increase in the share of gram-negative bacilli, this analysis was not possible to be performed.

## 4. Materials and Methods

We performed a retrospective study using clinical and laboratory data concerning bacteremia etiology and blood culture contamination over a 10-month period (March–December 2020) at the National Institute of Infectious Diseases “Prof. Dr. Matei Balș”, Bucharest, Romania. It is a monospeciality hospital, with 12 Infectious Diseases departments (more than 400 beds) and an intensive care unit (with 20 beds). During 2020, the hospital was designated exclusively for the treatment of COVID-19 patients. The results were compared with existing data on blood cultures performed for a 10-month period in the pre-COVID-19 era (March–December 2016).

The “infectious episode with blood cultures sampled” was defined as the presence of signs and symptoms suggestive of an infectious pathology that can occur once or several times during one hospitalization, in which blood cultures were collected.

For each infectious episode, we analyzed the blood cultures results, as seen in Figure 1. The isolated microorganisms were considered true etiological agents or a contaminant following the analysis carried out by two authors based on the following criteria. A microorganism isolated from blood cultures was considered a contaminant if it was a well-recognized less pathogen bacterium (coagulase-negative staphylococci, *Corynebacterium* species, *Bacillus* species other than *anthracis*, and *P. acnes*) and if it was isolated from a single blood culture bottle and clinical and biological data excluded a systemic infection.

### Statistical Analysis

Clinical and laboratory data were extracted from the database of the Bacteriology laboratory of the National Institute of Infectious Diseases “Prof. Dr. Matei Balș” and from its informatic system, were subsequently entered into a database and analyzed using the Fisher exact test and a two-tailed Wilcoxon signed-rank test. Epi Info version 7.0 statistical analysis software was used.

The results were compared with data identified in the pre-pandemic period (March–December 2016) and with published data from other hospitals.

## 5. Conclusions

The collection of a low number of blood cultures continues to be a practice in our clinic and an impediment in establishing a complete diagnosis.

In order to improve the usefulness of blood cultures as a diagnostic method, we must collect a larger number of blood cultures (a minimum two sets per patient with blood culture collection indications) before initiating antimicrobial therapy and following the collection protocol, in order to decrease the contamination risk. We should also include our nurses in specific blood culture collection programs and we should apply stricter rules when collecting blood cultures. This is important because many times the contaminations can be considered real positive, which generates the administration of an unjustified antibiotic treatment and the extension of the hospitalization period.

## Figures and Tables

**Figure 1 antibiotics-11-01694-f001:**
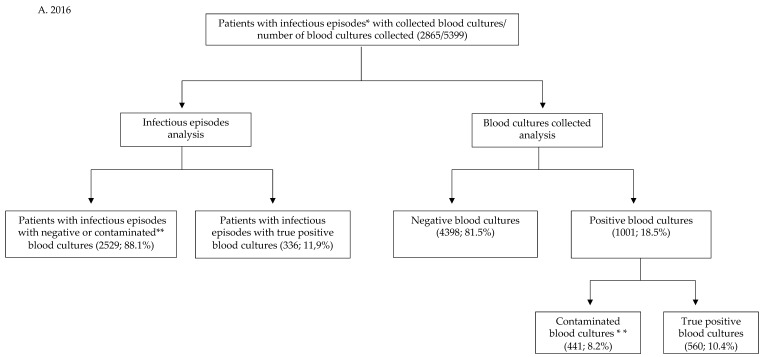
Distribution of patients into groups related to the blood culture results: (**A**) (2016) and (**B**) (2020).

**Table 1 antibiotics-11-01694-t001:** Pathogens isolated from blood cultures.

	Number of Isolates		Difference 2020 vs. 2016
Species	2016	2020	
*Escherichia coli*	95 (28.3%)	4 (4.8%)	***p* < 0.0001**
*Staphylococcus aureus*	71 (21.1%)	9 (10.7%)	***p* = 0.029**
*Klebsiella pneumoniae*	35 (10.4%)	16 (19%)	***p* = 0.031**
*Enterobacterales* (other)	24 (7.1%)	3 (3.6%)	*p* = 0.241
*Enterococcus* spp.	20 (5.9%)	6 (7.1%)	*p* = 0.682
*Streptococcus pneumoniae*	16 (4.8%)	1 (1.2%)	*p* = 0.136
*Acinetobacter baumannii*	15 (4.5%)	22 (26.2%)	***p* < 0.0001**
*Pseudomonas aeruginosa*	9 (2.7%)	6 (7.1%)	***p* = 0.052**
*Candida* spp.	9 (2.7%)	5 (5.9%)	*p* = 0.145
*Stenotrophomonas maltophilia*	1 (0.3%)	2 (2.4%)	***p* = 0.042**
Others	41 (12.2%)	10 (11.9%)	
TOTAL	336	84	

**Table 2 antibiotics-11-01694-t002:** Blood culture use in 2020 compared to 2016—hospital and medical unit type changes.

Parameter	2016 (n; %)	2020 (n; %)	Difference 2020 vs. 2016
Infectious episodes for which blood cultures were collected	2865	782	
Infectious episodes per medical unit type Infectious diseases ICU	2526 (88.2%) 339 (11.8%)	562 (71.9%) 220 (28.1%)	**z = 11.22 *p* < 0.0001**
Days of hospitalization	140,733	88,767	
Total number of blood cultures collected	5399	1274	
Blood cultures/1000 patient days	38.36	14.35	**z = 33.36 *p* < 0.00001**
Blood cultures collected per medical unit type Infectious diseases ICU	4850 (89.8%) 549 (10.2%)	831 (65.2%) 443 (34.8%)	**z = −22.13 *p* < 0.00001**
Blood culture sets/infectious episode	1.91 ± 1.4	1.67 ± 1.16	**t = −4.50** ***p* < 0.0001**
Blood culture sets per infectious episode and per medical unit type Infectious diseases ICU	1.92 ± 1.16 1.62 ± 1.01	1.48 ± 0.84 2.01 ± 1.26	**t = −10.47, *p* < 0.0001** **t = 5.41, *p* < 0.0001**
Infectious episodes with only one blood culture	1475 (51.48%)	489 (62.53%)	**z = −5.49 *p* < 0.0001**
Infectious episodes with only one blood culture per medical unit type Infectious diseases ICU	1257 (49.8%) 218 (64.3%)	378 (67.3%) 111 (50.5%)	**z = −7.52 *p* < 0.0001** **z = 3.25 *p* = 0.0011**

ICU = intensive care unit.

**Table 3 antibiotics-11-01694-t003:** Positive blood culture isolates, true positives, and contaminants: 2020 versus 2016.

Parameter	2016	2020	Difference 2020 vs. 2016
Contaminated blood cultures (blood culture contamination rate)	441 (8.2%)	149 (11.7%)	**z = −3.99 *p* = 0.00006**
Contaminants - Coagulase-negative staphylococci - *Corynebacterium* spp. - Propionibacterium acnes - Others	71.7% 7.3% 0.9% 20.1%	91.3% 1.3% 4.7% 2.7%	***p* < 0.0001**
Infectious episodes with true positive blood cultures (rate of infectious episodes with true positive blood cultures/infectious episodes with blood cultures sampled)	336 (11.9%)	84 (11%)	z = 0.71 *p* = 0.479
Infectious episodes with true positive blood cultures/1000 patient days	2.39	0.95	**z = 7.87 *p* < 0.00001**
Infectious episodes with true positive blood cultures/medical unit ICU Infectious diseases	36 (10.7%) 300 (89.3%)	53 (63.1%) 31(36.9%)	**z = 10.51, *p* < 0.00001**

ICU = intensive care unit.

## Data Availability

Not applicable.

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
