# Peer review of "Changes in Use of Blood Cultures in a COVID-19-Dedicated Tertiary Hospital"

_antibiotics, 2022, doi:10.3390/antibiotics11121694_

Round 1
Reviewer 1 Report
It was a pleasure to read this manuscript, many congratulations to the authors for their work and dedication in sharing local data.
It is a well written script, with clear data and possible interpretation for their results.
The only comment I have is pedantic in nature - line 29 delete "proper", insert "appropriate".
Otherwise I recommend publishing the manuscript in its current form.
With very best wishes
Author Response
Response to Reviewer 1 Comments
We thank the reviewer for the careful review of the submitted manuscript and for the comment.
Point 1: The only comment I have is pedantic in nature - line 29 delete "proper", insert "appropriate".
Response 1: We have made the suggested text change.
Reviewer 2 Report
In general, I find the topic and manuscript interesting; however, the practices of this institution seem so aberrant that it is difficult to draw any firm conclusions on how the findings may be relevant / extrapolated to other institutions. To make a more compelling manuscript, perhaps a different perspective should be taken.
Abstract:
The abstract is vague and needs clarification. For example, in lines 18 – 22, it is difficult to discern which data are from pre-pandemic and which are from post-pandemic periods.
Results:
What is an ‘infectious episode?’ This terminology should be clarified or defined.
Can you give data for what percentage / how many cultures were collected from various units (ICU, ED or otherwise)? I see that this is noted in Table 2, but it includes ‘Infectious Diseases’ and ‘ICU.’ Were there no other units?
As noted in my comments on ‘materials and methods,’ a flow diagram of how the determination was made for pathogen vs contaminant would be helpful for the reader.
Table 3: “Infections episodes with positive blood cultures” should be clarified so that the reader can easily determine that this is only true positives vs all positives. It may be helpful to include the number of true positives and contaminants in this table.
Discussion:
It would be reasonable to include the patient volume in this discussion. I do realize that you include patient days, but this could be misleading as, during COVID, length of stay likely increased and this may be why you see such an increase in the hospital-acquired gram negatives (eg, Acinetobacter spp). It would be helpful to normalize this for the total number of patient visits as well.
Line 168 – This sentence is unclear. Does this mean that the clinicians were ordering one set instead of two? Is this an option for clinicians? If so, why is this the case? In cases where only one set was collected, in what percentage of these was only one set ordered vs the phlebotomy or nursing team was only able to collect one set?
Paragraph beginning line 181 – The total length of stay for these patients prior to the positive blood culture would be helpful, especially as compared to the pre-pandemic length of stay. It may help clarify the following paragraph when speaking about the pathogens. I would anticipate that those in whom a hospital-acquired organism was collected had been in the hospital longer.
Another factor to consider in the discussion: Because the contamination rate is roughly equal to the true positive rate, from the clinician perspective, half of all positive cultures are contaminants. This places difficult decisions on the clinicians and has the possibility of resulting in neglecting to treat a true pathogen because the rate of contamination is so high.
Line 224 – please clarify the number of patients for whom all data are not available.
Materials and Methods:
Were all blood cultures during these periods included in the study? A study flow diagram might be helpful to depict how many (if any) cases were excluded from analysis and why they may have been excluded. Also, it would be helpful to include specific criteria and the process used for determination of contaminant. This process should be reproducible and structured to prevent inter-observer variability in determination of a true pathogen vs contamination. Were the same criteria for determination of contamination applied to the pre- and post-pandemic cohorts?
Again, a flow diagram depicting how many cultures were positive and how many determined to be contamination and why would be helpful? In this way, it may be valuable for the reader to determine if there were differences in these periods in the type of contamination and reason for occurrence. For example, were there equal numbers of common commensals recovered during these periods, or did the ‘clinical and biological data’ account for more determination of one period over another?
Conclusions:
A statement on the impact that contamination has on clinical care and postulated areas for improving the quality of collection to prevent contamination should be discussed. Collecting for cultures with the stated contamination rate may only compound the problem.
Author Response
Response to Reviewer 2 Comments
We thank the reviewer for the attention with which he/she analyzed the submitted manuscript and for the comments made. We agreed with a good part of them and modified the text of the manuscript accordingly. Another part of the observations was determined by the particularities of medical care in Romania, especially related to the existence of monospeciality hospitals for Infectious Diseases.
Point 1:
Abstract:
The abstract is vague and needs clarification. For example, in lines 18 – 22, it is difficult to discern which data are from pre-pandemic and which are from post-pandemic periods.
Response 1: We clarified which data is from 2020 and which is form 2016.
Point 2:
Results:
What is an ‘infectious episode?’ This terminology should be clarified or defined.
Response 2: We introduced the definition of "infectious episode with collected blood cultures" in the Methodology.
Point 3:
Results:
Can you give data for what percentage / how many cultures were collected from various units (ICU, ED or otherwise)? I see that this is noted in Table 2, but it includes ‘Infectious Diseases’ and ‘ICU.’ Were there no other units?
Response 3: Regarding the clinical departments in which the patients were admitted, we specified that our hospital has only Infectious Diseases departments and an Intensive Care Unit.
Point 4:
Results:
As noted in my comments on ‘materials and methods,’ a flow diagram of how the determination was made for pathogen vs contaminant would be helpful for the reader.
Response 4: We have added a diagram that allows patients with collected blood cultures to be classified into the three categories: with negative, contaminated or true positive blood cultures.
Point 5:
Results:
Table 3: “Infections episodes with positive blood cultures” should be clarified so that the reader can easily determine that this is only true positives vs all positives. It may be helpful to include the number of true positives and contaminants in this table.
Response 5: We made the proposed clarifications for Table 3 related to the analyzed categories; the number of true positives and contaminants already existed in the table.
Point 6:
Discussion:
It would be reasonable to include the patient volume in this discussion. I do realize that you include patient days, but this could be misleading as, during COVID, length of stay likely increased and this may be why you see such an increase in the hospital-acquired gram negatives (eg, Acinetobacter spp). It would be helpful to normalize this for the total number of patient visits as well.
Response 6: We did not include the total number of hospitalized patients in the two periods because we considered much more important the number of patients with blood cultures collected and the number of days of hospitalization.
Point 7:
Discussion:
Line 168 – This sentence is unclear. Does this mean that the clinicians were ordering one set instead of two? Is this an option for clinicians? If so, why is this the case? In cases where only one set was collected, in what percentage of these was only one set ordered vs the phlebotomy or nursing team was only able to collect one set?
Response 7:
We tried to clarify the discussions from row 168, related to the constant rate of real positive blood cultures.
Related to the number of blood cultures collected, there is this problem of collecting a single set, which worsened during the pandemic. Both proposed explanations are real: both the difficulty of collecting a second set, and the fact that sometimes only one set of blood cultures was requested. Our aim was to show that the size of this phenomenon, of the unique set of blood cultures, was amplified during the pandemic.
Related to the collection by the dedicated team, in our hospital, as in all hospitals in Romania such a team does not exist, mainly due to the shortage of nurses (who cannot be dedicated to a single type of activity).
Point 8:
Discussion:
Paragraph beginning line 181 – The total length of stay for these patients prior to the positive blood culture would be helpful, especially as compared to the pre-pandemic length of stay. It may help clarify the following paragraph when speaking about the pathogens. I would anticipate that those in whom a hospital-acquired organism was collected had been in the hospital longer.
Response 8: We agree with the hypothesis put forward by the reviewer, which we were also discussing in relation to the change in the etiological picture of bacteremias in 2020 compared to 2016. The analysis of the length of hospitalization until the time of collection of the real positive blood culture is, however, difficult to be appreciated, because a proportion of the patients were hospitalized by transfer from other hospitals, where they were previously hospitalized for variable periods of time (our hospital is a tertiary center).
Point 9:
Discussion:
Another factor to consider in the discussion: Because the contamination rate is roughly equal to the true positive rate, from the clinician perspective, half of all positive cultures are contaminants. This places difficult decisions on the clinicians and has the possibility of resulting in neglecting to treat a true pathogen because the rate of contamination is so high.
Response 9: The hypothesis regarding the difficulty that can sometimes arise regarding the identified bacteria is correct: real pathogen or contaminant. However, the risk evoked by the reviewer, that of leaving a real pathogen untreated, is extremely low, given that in general (and especially during the pandemic) both in Romania and in other parts of the world, antibiotics have been used in excess.
Point 10:
Discussion:
Line 224 – please clarify the number of patients for whom all data are not available.
Response 10: We have made the requested modification - by omitting the respective paragraph (line 222-225).
Point 11:
Materials and Methods:
Were all blood cultures during these periods included in the study? A study flow diagram might be helpful to depict how many (if any) cases were excluded from analysis and why they may have been excluded. Also, it would be helpful to include specific criteria and the process used for determination of contaminant. This process should be reproducible and structured to prevent inter-observer variability in determination of a true pathogen vs contamination. Were the same criteria for determination of contamination applied to the pre- and post-pandemic cohorts?
Again, a flow diagram depicting how many cultures were positive and how many determined to be contamination and why would be helpful? In this way, it may be valuable for the reader to determine if there were differences in these periods in the type of contamination and reason for occurrence. For example, were there equal numbers of common commensals recovered during these periods, or did the ‘clinical and biological data’ account for more determination of one period over another?
Response 11: Yes, the same criteria were used in 2016 and 2020 to define a contaminated blood culture. We have included the requested chart, which also shows the criteria used to define contamination.
Criteria were cumulative to assert contamination. This might indeed introduce a small number of contaminants among the true positives, but any way of differentiation has its inaccuracies and leaves an area of uncertainty.
Point 12:
Conclusions:
A statement on the impact that contamination has on clinical care and postulated areas for improving the quality of collection to prevent contamination should be discussed. Collecting for cultures with the stated contamination rate may only compound the problem.
Response 12: We have implemented the suggested changes.
Reviewer 3 Report
The manuscript did not give a clear answer to the aim.
Material (4.) has been very poorly discussed. Also, it is in the wrong place in the manuscript (after chapter 3. Discussion).
The lack of a definitive distinction between pathogens and microorganisms contaminating the samples disqualifies the authors' results based on the analysis of such material.
There is no reference in the text to Table 1.
Tables 2 and 3 have titles inadequate to their content; they also lack legends, explanation of abbreviations, for example ICU.
Summing up, the authors rightly point out the limitations in the work regarding the research material and access to data, which in turn allowed for a proper comparative analysis and achievement of the assumed research aims.
Author Response
Response to Reviewer 3 Comments
We thank the reviewer for the attention with which he/she analyzed the submitted manuscript and for the comments made. We agreed with a good part of them and modified the text of the manuscript accordingly.
Point 1: Material (4.) has been very poorly discussed. Also, it is in the wrong place in the manuscript (after chapter 3. Discussion).
Response 1:
Related to the differentiation between contaminants and true positives, the criteria existed and are now more explicit in the Materials and Methods section.
We used the MDPI Microsoft Word template for writing the manuscript.
Point 2: There is no reference in the text to Table 1.
Response 2: We have corrected and there is now reference to Table 1 in the text.
Point 3: Tables 2 and 3 have titles inadequate to their content; they also lack legends, explanation of abbreviations, for example ICU.
Response 3: We modified the titles of tables 2 and 3 and explained the abbreviation used.
Round 2
Reviewer 2 Report
Overall my concerns have been acceptably addressed.
Reviewer 3 Report
Thank you for considering my comments. I accept the changes made.